# Grain Boundary Engineering in 3D Porous Silver Electrocatalysts for Enhanced CO_2_-to-CO Conversion

**DOI:** 10.3390/molecules30173475

**Published:** 2025-08-24

**Authors:** Xiaoqian Xu, Song Yang, Yixiang Wang, Ying Chen, Assa Aravindh Sasikala Devi, Feng Hu

**Affiliations:** 1Jiangsu Key Laboratory of Electrochemical Energy Storage Technologies, College of Materials Science and Technology, Nanjing University of Aeronautics and Astronautics, Nanjing 210016, China; 2Research Unit of Sustainable Chemistry, Faculty of Technology, University of Oulu, 90014 Oulu, Finland; 3Department of Physics, Durham University, Lower Mountjoy, South Rd, Durham DH1 3LE, UK

**Keywords:** silver-based catalyst, nanoparticle attachment, electrocatalysis, carbon dioxide reduction, CO formation

## Abstract

Silver is a promising electrocatalyst for electrochemical CO_2_ reduction reaction owing to its high selectivity and efficiency for CO production. However, it still faces a fundamental trade-off between reaction activity and stability. Here, we developed a three-dimensional coral-like porous silver (CP-Ag) catalyst through seed-assisted nanoparticle attachment synthesis, which creates a unique architecture featuring interconnected pores and stable grain boundaries (GBs) between constituent Ag nanoparticles (Ag NPs). Compared to normal Ag NPs, CP-Ag demonstrates superior catalytic performance, maintaining >90% Faradaic efficiency (FE) for CO across a wide potential range (−0.6 to −1.0 V vs. RHE) while achieving 2-times higher current density. Importantly, CP-Ag demonstrated an impressive long-term stability by sustaining nearly 90% FE for CO approximately 40 h at a current density of −50 mA cm^−2^ in a flow cell. The enhanced catalytic performance arises from three factors: (1) the three-dimensional coral-like morphology increases accessible active sites and promotes charge transfer efficiency; (2) stable GBs between interconnected nanoparticles increase reaction activity; (3) more moderate binding on Ag (100) preferentially promotes *CO intermediate formation. Our findings highlight the importance of simultaneously engineering both morphological and crystallographic features to optimize silver catalysts for CO_2_ conversion.

## 1. Introduction

Electrochemical CO_2_ reduction (CO_2_RR) has emerged as a highly promising approach for converting CO_2_ into value-added chemical products, particularly CO. As a crucial intermediate in chemical manufacturing, CO production via CO_2_RR exhibits distinct advantages, including relatively simple reaction pathways and lower overpotentials in most catalytic systems, thereby significantly improving the energy efficiency of the CO_2_RR process [1,2,3,4,5]. The electrochemical reduction of CO_2_ to CO typically follows a well-established reaction pathway involving multiple sequential steps, as comprehensively outlined in the recent literature [6]. This process begins with the adsorption of reactants onto the electrocatalyst surface, followed by a critical proton-coupled electron transfer (PCET) event that drives chemical conversion, and concludes with the desorption of products to regenerate active sites. Within this general framework, a well-defined pathway for CO formation from CO_2_ proceeds via a sequence of surface-bound intermediates: *COOH, *CO, and *H, where * signifies an adsorbed species or catalyst surface vacancy. This mechanism specifically involves four key mini-steps: First, an adsorbed CO_2_ molecule undergoes an initial electron transfer, forming a surface-bound *CO_2_^−^ intermediate. This highly reactive species is pivotal for the subsequent steps. Second, the resulting *CO_2_^−^ intermediate is then protonated by H^+^, largely from H_2_O, leading to the formation of a *COOH intermediate. This protonation is crucial for stabilizing the intermediate and directing the reaction pathway. Third, the *COOH intermediate subsequently undergoes a critical dehydration step directly on the catalyst surface. This intramolecular elimination of water efficiently generates the pivotal surface-bound *CO intermediate. Finally, to complete the catalytic cycle and obtain the desired product, the newly formed *CO intermediate desorbs from the catalyst surface, releasing the CO product and regenerating the active sites for continuous operation [7,8].

As is well-known, Ag and Au noble metal catalysts exhibit exceptional electrocatalytic activity for CO_2_-to-CO conversion, opening up significant prospects for practical applications. Their high efficiency in this crucial reduction reaction contributes to addressing the escalating carbon emission dilemma [9,10,11,12]. Among these, Ag-based catalysts have garnered particular attention due to their cost advantage over Au, while demonstrating comparable catalytic performance. This positions Ag as a more economically viable and sustainable option for large-scale implementation. However, conventional Ag catalysts face two critical limitations. Firstly, achieving acceptable reaction rates necessitates high overpotentials, severely hindering energy efficiency. Such high overpotential requirements lead to increased energy consumption, impacting overall economic viability and sustainability. Secondly, at these elevated potentials, the competing hydrogen evolution reaction (HER) becomes increasingly dominant, significantly compromising CO selectivity. The enhanced HER not only diminishes the yield of the target product, CO, but also complicates separation and purification processes [13,14,15,16]. These limitations stem from the inherent stability of CO_2_ molecules and suboptimal binding energies of reaction intermediates on conventional catalyst surfaces, leading to inefficient reaction pathways and undesirable side reactions. Therefore, designing and developing novel Ag catalysts to overcome these intrinsic barriers remains a crucial challenge in the current CO_2_ reduction field.

Nanoporous architectures overcome these challenges through two synergistic mechanisms: (1) hierarchical pore networks that markedly increase the density of catalytically active sites, and (2) engineered surface electronic states combined with optimized mass transport properties that effectively suppress the competing hydrogen evolution reaction (HER) [17,18,19]. Lu et al. [20] prepared a nanoporous silver electrocatalyst, which can reduce CO_2_ to CO with a selectivity of about 92% at a moderate overpotential of 0.50 V, and its current density is more than 3000 times higher than that of a polycrystalline silver catalyst. The high performance was attributed to the larger electrochemical surface area and highly active curved surface inherent to the porous structure. Beyond morphological control, crystallographic engineering emerges as a powerful strategy for optimizing Ag catalysts [21,22,23,24]. A higher grain boundaries (GBs) density within the catalysts enhances CO_2_-to-CO conversion and suppressed HER, owing to GBs-induced atomic disorder that lowers activation barriers and stabilizes key *COOH intermediates [25]. These findings establish that synergistic combination of nanoscale porosity and crystallographic control represents the most promising path forward for developing practical CO_2_RR catalysts, achieving both the high activity required for industrial relevance and the selectivity needed for energy-efficient operation.

Herein, we developed a three-dimensional coral-like porous silver (CP-Ag) catalyst via a seed-assisted nanoparticle assembly approach, creating an interconnected porous structure with abundant GBs. The CP-Ag catalyst exhibits exceptional CO_2_ reduction performance, achieving 96% FE for CO at −1.0 V vs. RHE, doubling conventional Ag NPs’ performance. Notably, CP-Ag maintained nearly 90% FE for CO approximately 40 h of continuous operation at a current density of −50 mA cm^−2^ in a flow cell, showcasing exceptional long-term stability. Theoretical calculation demonstrates that the performance improvement arises from the 3D porous network’s increased active site availability and grain boundaries’ dual role in lowering activation energies and stabilizing *COOH intermediates. Our results demonstrate that simultaneous control over nanoscale morphology and crystallographic orientation can significantly boost silver catalysts’ CO_2_-to-CO conversion, providing a design strategy for high-performance electrocatalysts through combined structural and facet engineering.

## 2. Results and Discussion

Three-dimensional porous silver nanostructures were constructed via a seed-mediated morphological evolution from discrete Ag NPs to an interconnected network architecture, while maintaining comparable particle dimensions (Appendix A). The structural evolution was accomplished through a precisely controlled synthesis protocol involving a reaction mixture comprising silver nitrate, trisodium citrate (acting as stabilizer), and ascorbic acid (serving as reducing agent). Subsequent introduction of sodium borohydride initiated nanoparticle interconnection, ultimately yielding the three-dimensional porous architecture (Figure 1a). Scanning electron microscopy (SEM) analysis revealed a morphological transition from dispersed AgNPs to an interconnected porous structure, denoted as CP-Ag (coral-like porous Ag) (Figure 1b and Appendix A). The high-resolution Transmission electron microscopy (HRTEM) analysis (Figure 1c–e) further reveals the formation of coral-like architectures with distinct grain boundaries between crystalline domains, where lattice spacings of 0.22 nm and 0.25 nm correspond to the (100) and (111) planes of face-centered cubic silver, respectively [26,27,28]. The corresponding elemental mapping (Figure 1f) confirms the uniform elemental dispersion of Ag and O. All these morphology results manifest the transition from dispersed AgNPs to interconnected porous CP-Ag.

The phase structure was then studied by the X-ray diffraction (XRD) and X-ray photoelectron spectroscopy (XPS) analysis. XRD analysis definitively confirmed the face-centered cubic (fcc) crystalline structure for both the initial AgNP precursors and the final interconnected CP-Ag architectures (Figure 2a) [29]. There are 2θ characteristic peaks at 38.1°, 44.3°, 64.4°, and 77.5°, corresponding to the (111), (200), (220), and (311) crystal planes, respectively. Significantly, no new peaks or substantial shifts in existing peak positions were detected in the CP-Ag sample following sodium borohydride treatment and subsequent morphological evolution. This remarkable finding, coupled with the maintenance of identical relative peak intensities, provides compelling evidence that the underlying crystalline structure of silver is robustly preserved despite the morphological changes (e.g., interconnection and pore formation) observed via TEM. This structural integrity is crucial for maintaining the intrinsic properties of the silver material. XPS probed the electronic structure evolution during the morphological transformation (Figure 2b). The high-resolution Ag 3d spectra of AgNPs exhibited two distinct peaks at 374.10 eV and 368.10 eV, corresponding to the 3d_3/2_ and 3d_5/2_ spin–orbit components, respectively. After sodium borohydride treatment, CP-Ag showed Ag 3d_3/2_ and 3d_5/2_ peaks at 374.00 eV and 367.96 eV, respectively. Analysis of AgNPs and CP-Ag revealed shifts in their Ag 3d3/2 and 3d5/2 peaks, suggesting potential surface oxidation. However, the electronic valence state of Ag remains essentially unchanged before and after the reaction, with both forms predominantly existing as metallic silver. This indicates that while partial surface oxidation may occur, the bulk Ag retains its metallic character. Collectively, these comprehensive morphological and crystallographic results compellingly demonstrate the successful and controlled transition from discrete Ag NPs to a highly interconnected and porous CP-Ag nanostructure, paving the way for potential applications leveraging its unique structural attributes.

To quantify the exposed low-index facets of silver, we employed cyclic voltammetry (CV) in CO_2_-saturated 0.1 M KHCO_3_ electrolyte (50 mV/s scan rate). This methodology is particularly powerful as the adsorption behavior of HCO_3_^−^ (bicarbonate ions) is well-established to be facet-dependent and occurs at distinct potential windows for different silver crystal planes [30,31]. Both CP-Ag and AgNP electrodes displayed three characteristic redox peaks (Figure 2c,d): Ag (110) (−1.15 to −0.89 V), Ag (100) (−0.67 to −0.43 V), and Ag (111) (−0.43 to −0.35 V vs. Ag/AgCl). Structural analysis revealed that the (100) facet consistently exhibited the highest peak intensity among all identified crystal planes specifically in the CP-Ag catalyst. This remarkable signal enhancement correlated well with charge distribution measurements, which demonstrated that the (100) plane possessed the highest charge density. These observations strongly indicate that the (100) facet serves as the predominant active site in the CP-Ag catalyst, where its unique atomic configuration and electronic properties synergistically contribute to the observed catalytic performance. In addition, Table 1 further demonstrated the electrochemical properties of different crystal planes of CP-Ag and AgNPs. The capacitance and proportion of the (100) plane of CP-Ag is the largest. The strategic exposure and stabilization of the highly reactive Ag (100) facets, achieved via our controlled synthesis, fundamentally optimizes CP-Ag as an exceptional catalyst. This enhancement is directly correlated with the predominant presence of Ag (100) facets, which exhibit a strong propensity for the preferential adsorption of critical reaction ions and intermediates. To further elucidate the origin of this enhanced activity, CV curves were obtained at different scanning speeds (10, 20, 40, 60, 80, 100, 120, 140 mV s^−1^) (Appendix A), and the electrochemical active area of CP-Ag and AgNP was evaluated by calculating the electrochemical surface area (ECSA) and double-layer capacitance (Cdl). ECSA measurements revealed that CP-Ag (1.61 mF cm^−2^) possesses a 18% larger double-layer capacitance (Cdl) than AgNPs (1.36 mF cm^−2^) (Figure 2e,f), attributable to its three-dimensional porous architecture [32]. However, the facet-dependent CV results clearly demonstrate that the improved catalytic performance stems primarily from the intrinsic catalytic properties of the dominant (100) facets, rather than being solely a consequence of increased surface area.

The electrocatalytic performance of CP-Ag was evaluated in a CO_2_-saturated 0.1 M KHCO_3_ electrolyte (pH = 6.8) using an H-type electrolyzer, with Ag foam and AgNPs as controls. Figure 3a shows the linear sweep voltammetry (LSV) curves of CP-Ag and AgNP in CO_2_-saturated electrolytes with a scan rate of 10 mV s^−1^. The LSV revealed that CP-Ag exhibited significantly enhanced cathodic kinetics, achieving a current density of −16.41 mA cm^−2^ at −1.7 V vs. RHE, approximately twice that of AgNPs (−8.56 mA cm^−2^) (Figure 3a). Product analysis revealed outstanding CO selectivity for the CP-Ag catalyst, which maintained FE_CO_ exceeding 90% across an extensive potential range from −0.6 to −1.0 V vs. RHE, peaking at an exceptional 96.6% FE at −1.0 V vs. RHE and no detectable liquid products (Appendix A). In stark contrast, AgNPs and Ag foam exhibited inferior performance (Figure 3b–d and Appendix A). More importantly, electrochemical measurements showed that the CO partial current density of CP-Ag doubled that of AgNPs, unambiguously confirming its superior CO_2_RR activity. This remarkable performance enhancement can be attributed to the unique structural characteristics of CP-Ag, including its optimized crystal facet exposure and favorable electronic structure, which collectively promote CO_2_ activation and CO desorption.

To reveal the mechanism behind the performance enhancement, the reaction kinetics of the catalyst in the electrocatalytic reduction of CO_2_ were studied. Building upon these performance results, Tafel analysis further revealed a significantly lower Tafel slope of 147 mV dec^−1^ for CP-Ag compared to 230 mV dec^−1^ for AgNP. This suggests faster reaction kinetics and higher catalytic activity of CP-Ag (Figure 3f) [33]. In a further comparison of the conductivity of the CP-Ag, AgNP and Ag foam catalysts, the electrochemical impedance spectra (EIS) of CP-Ag, AgNP and Ag foam were determined in a CO_2_-saturated 0.1 M KHCO_3_ electrolyte. As shown in Appendix A, the CP-Ag exhibited a smaller equivalent series resistance than that of AgNP and Ag foam, illustrating faster catalytic kinetics. As is well known, the flow cell configuration offers distinct advantages for CO_2_RR by enabling high current densities through enhanced mass transport and efficient gas–liquid-solid interfaces. In our tests, the CP-Ag sustained approximately 90% FE for CO about 40 h of continuous operation at a current density of −50 mA cm^−2^ in a flow cell, demonstrating outstanding long-term stability (Figure 3g). The combined structural and electronic effects enable CP-Ag to simultaneously suppress the competing hydrogen evolution reaction while promoting CO_2_-to-CO conversion. Additionally, the CP-Ag catalyst exhibits superior CO_2_RR performance among the Ag-based catalysts reported in previous studies (Appendix A). These results collectively establish CP-Ag as a highly efficient and tunable electrocatalyst for the reduction of CO_2_ to CO, offering significant promise for electrocatalytic applications.

To evidence the origin of electrocatalysis activity, we performed density functional theory (DFT) to explain the reaction kinetics. The *COOH intermediate is an important intermediate towards the production of CO products. Therefore, we calculated the adsorption energies of the key intermediate *COOH (E_ads_(*COOH)) using surfaces of Ag (100) and Ag (111). The configurations with adsorbed intermediates are given in Figure 4a,b. The adsorption energies corresponding to the adsorption patterns for Ag (100) and Ag (111) are listed in Appendix A. DFT calculations reveal distinct facet-dependent adsorption characteristics for *COOH on silver surfaces, with binding energies of −1.57 eV on Ag (100) and −1.62 eV on Ag (111). This energetic difference leads to different adsorption energy: the stronger adsorption on Ag (111) likely elevates the activation barrier for *COOH dissociation, while the more moderate binding on Ag (100) preferentially promotes the desorption of *CO intermediate, thereby enhancing the rate of CO_2_RR [34,35,36].

## 3. Experimental Section

### 3.1. Materials

Silver foam (99.9%), Silver nitrate (AgNO_3_), sodium borohydride (NaBH_4_), KHCO_3_ (99.9%), CO_2_ (99.998%), Ar (99.998%) and N_2_ (99.99%) were provided by Nanjing Analytical (Nanjing, China). Carbon Paper (CP, 19 × 19 cm) and Nafion N-115 membrane (0.180 mm thick, ≥0.90 meg/g exchange capacity) were purchased from Nanjing Chemical Reagent (Nanjing, China). Gas diffusion layer (28BC of Freudenberg, Weinheim, Germany), membrane (Nafion 117, Chemours, Wilmington, DE, USA), Nafion (Aladdin, Shanghai, China, 5%), isopropyl alcohol (C_3_H_8_O, Maklin, Shanghai, China, AR 98%) were used for electrochemical CO_2_ measurements. Ultrapure water (18.2 MΩ·cm^−1^) was obtained from an ELGA ultrapure water purification system. All chemicals were used without further purification.

### 3.2. Synthesis of CP-Ag Catalysts

With stirring at room temperature, in a 20 mL vial, 11 mL of the mixed solution containing 0.11 mM silver nitrate and 2.05 mM trisodium citrate was prepared, following by the addition of 100 μL 5 mM sodium borohydride solution. After stirring for 10 min, the solution was left for aging at room temperature for 5 h to obtain Ag seed solution. Under stirring at room temperature, 0.2 mL of Ag seed solution, 7.5 mL of trisodium citrate (30 mM) and 3 mL of silver nitrate (50 mM) were sequentially added to a 100 mL beaker, and the solution turned milky white. Then 62.5 mL of ascorbic acid (1 mM) was slowly added to the mixed solution. After the dropwise addition, 12 mL of sodium borohydride (1 M) was added to the mixed solution at one time, and the solution reacted violently with the generation of bubbles. The reaction continued for 5 min, the stirring was stopped, the solution was allowed to stand in the dark for 1 h, then the solution was removed to obtain a precipitate and the precipitate was freeze-dried to obtain a product designated as CP-Ag.

### 3.3. Synthesis of Ag NPs

Under stirring at room temperature, 0.2 mL of Ag seed, 7.5 mL of trisodium citrate (30 mM) and 3 mL of silver nitrate (50 mM) were sequentially added to a 100 mL beaker, and the solution turned milky white. Then 62.5 mL of ascorbic acid (1 mM) was slowly added to the mixed solution. After the dropwise addition, the mixed solution was centrifuged and freeze-dried to obtain silver nanoparticles (Ag NPs).

### 3.4. Materials Characterization

Scanning electron microscopy (SEM, Regulus 8100 apparatus) and transmission electron microscopy (TEM, FEI Tecnai G2 F20, Hillsboro, OR, USA) equipped with energy dispersive spectrometer (EDS) were used to analyze the sample morphology and element distribution. X-ray diffraction (XRD, Bruker D8 ADVANCE, Karlsruhe, Germany, Cu Kα) patterns were used to study the catalyst constitution and phase. X-ray photoelectron spectroscopy (XPS, Escalab 250Xi electron spectrometer, Thermo Fisher Scientific, Altrincham, UK) results were collected to analyze the catalyst element valences.

### 3.5. Electrochemical CO_2_ Reduction

All electrochemical measurements were conducted under ambient temperature and pressure conditions using a CS2350H electrochemical workstation (Wuhan Corrtest Instruments Corp., Ltd., Wuhan, China) with either an H-type cell or flow cell configuration. For H-cell measurements, a two-compartment electrolytic cell (cathodic and anodic chambers) separated by a Nafion 117 proton exchange membrane was employed. The cell was filled with 0.1 M KHCO_3_ electrolyte saturated with CO_2_, equipped with a KCl-saturated Ag/AgCl reference electrode in the cathodic compartment and a platinum sheet counter electrode in the anodic compartment. In flow cell experiments, both anolyte and catholyte were continuously recirculated through their respective compartments at a constant flow rate of 50 mL·min^−1^ using two identical peristaltic pumps (Kamoer LLS PLUS, Shanghai, China), while CO_2_ was introduced at a flow rate of 22 mL·min^−1^. The gaseous effluent from the cathodic chamber was monitored in real-time using online gas chromatography. The working electrode was prepared by dip-coating 100 μL of catalyst ink onto a 1 cm × 1 cm carbon cloth substrate (Toray TGP-H-060, Tokyo, Japan). The catalyst ink was formulated by dispersing 7 mg of catalyst with 50 μL Nafion solution in 950 μL isopropanol, followed by 30 min of sonication to achieve homogeneous dispersion.

Electrochemical characterization included linear sweep voltammetry (LSV) performed within a potential window of −0.6 to −2.4 V vs. Ag/AgCl at a scan rate of 50 mV·s^−1^. All current densities reported throughout this work were calculated based on geometric electrode area normalization. Potentials were measured against a KCl-saturated Ag/AgCl reference electrode and subsequently converted to the reversible hydrogen electrode (RHE) scale according to the Nernst equation: E(vs. RHE) = E(vs. Ag/AgCl) + 0.197 V + (0.0591 × pH), where E(vs. Ag/AgCl) represents the experimentally measured potential. Unless specifically noted, all potential values presented in both the main text and Appendix A are reported versus the RHE reference without iR compensation. This potential conversion was consistently applied to ensure standardized reporting of electrochemical data across all experiments.

### 3.6. Product Quantifications

Gas-phase products from the cathodic compartment were directly vented into a gas chromatograph (Fuli GC9790 II, Zhejiang Fuli Analytical Instrument Co., Ltd., Taizhou, China) using a flame ionization detector (FID) and a thermal conductivity detector (TCD) during the electroreduction tests and analyzed online. FID was used for CO quantification (as well as CH_4_, C_2_H_4_ and C_2_H_6_), while TCD was used for H_2_ quantification. All faradaic efficiencies reported were based on at least five different GC runs. High purity argon (99.999%) was used as the GC carrier gas. In all the CO_2_ electrolysis tests, only H_2_ and CO were the gas-phase products, and their faradaic efficiencies were calculated as follows:$$\mathrm{FE}\;=\;\frac{j_{x}}{j_{\mathrm{total}}}\;=\;\frac{n_{x}\;\text{×}\;z_{x}\;\text{×}\;v_{\mathrm{gas}}\;\text{×}\;F}{j_{\mathrm{total}}\;\text{×}\;V}\;\text{×}\;100\%$$
where *j*_x_ denotes the partial current density of the product, *j*_total_ represents the total current density, n_x_ indicates the amount of substance of a specific product (mol), z_x_ corresponds to the number of electrons transferred per mole of product generated, v_gas_ signifies the CO_2_ flow rate (m^3^ s^−1^), F is the Faraday constant (96,485 C mol^−1^), and V represents the injection volume (m^3^).

The electrochemical active surface area (ECSA) refers to the effective area participating in electrochemical reactions. Here, we employed the commonly used double-layer capacitance method to determine the ECSA. First, a non-faradaic potential window was selected, within which cyclic voltammetry (CV) tests were performed at different scan rates (CV curves at various scan rates are shown in Appendix A). The absolute difference between the anodic and cathodic currents at the midpoint potential of the CV curves was calculated, and half of this value was plotted on the y-axis, with the corresponding scan rate on the x-axis. The slope obtained from the linear fitting of this plot (Figure 2f) represents the double-layer capacitance. Subsequently, the ECSA values of the two samples were calculated using the formula:$$\mathrm{ECSA}\;=\;\frac{Cdl}{Cs}$$
where Cs is the specific capacitance, and a commonly cited value of 40 μF cm^−2^ was adopted from literature [37,38].

### 3.7. Computational Details

All DFT calculations were conducted using the Vienna ab initio simulation package (VASP, https://www.vasp.at/). The electronic exchange was described by a generalized gradient approximation method [34]. We constructed Ag (100) and Ag (111) oriented surfaces and performed optimization. These two surface orientations were chosen because they are the most catalytically active surfaces. Before constructing the surfaces, we first optimized the bulk structure of all materials, then created the surfaces and further performed ionic optimization. To generate the surfaces, we used the supercell slab technique, adding a sufficient vacuum layer along the Z-direction to avoid periodic interactions between images.

The adsorption energies (E_ads_) for *COOH was calculated by the following equation:$$E_{\mathrm{ads}}\;=\;E_{\mathrm{adsorbate}+\mathrm{surf}}\;-\;(E_{\mathrm{adsorbate}}\;+\;E_{\mathrm{surf}})$$
where E_adsorbate+surf_ is the total free energy of the surface with the functional attached to it, Eadsorbate is the single-point energy of the free standing functional, and Esurf is the total free energy of the pristine surface without the functional. With these definitions, a more negative E_ads_ reflects a stronger surface adsorption interaction. For *COOH is calculated as the energy of free standing *COOH molecules. The distance of the adsorbate after optimization from the surface is calculated in Angstrom units.

## 4. Conclusions

In this work, we demonstrated a rational synthesis of three-dimensional porous CP-Ag with abundant GBs through sodium borohydride-mediated transformation of seed-grown AgNPs. The CP-Ag catalyst exhibited exceptional CO_2_ reduction performance, achieving 96% FE for CO at −1.0 V vs. RHE, which is double that of the seed-grown Ag NPs, demonstrating its superior activity and selectivity. Significantly, CP-Ag demonstrated outstanding long-term stability by sustaining a nearly 90% FE for CO over about 40 h of continuous operation at a current density of −50 mA cm^−2^ in a flow cell. The enhanced CO_2_ reduction performance of the CP-Ag catalyst can be attributed to its unique coral-like architecture that increases active site density and improves electron transfer. DFT calculations reveal that facet-dependent *COOH adsorption energies influence the activation of key intermediates, with the moderate binding on Ag (100) favoring *CO formation and overall catalytic efficiency. These findings highlight the importance of rationally designing morphological and crystallographic features in electrocatalysts to attain high performance. Future efforts should focus on tailoring such nanostructures and facets to further optimize catalytic efficiency, demonstrating that the strategic engineering of microstructure and surface chemistry in silver-based catalysts holds significant potential for advancing CO_2_ conversion technologies and broadening the scope of catalytic material development in electrocatalysis research.

Entry for the Table of Contents

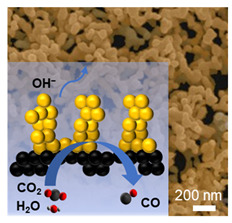



A 3D coral-like porous silver catalyst was synthesized with interconnected pores and stable grain boundaries between Ag NPs. It achieves over 90% Faradaic efficiency for CO across −0.6 to −1.0 V vs. RHE, and 40 h stability at −50 mA cm^−2^. The performance benefited from its porous structure for improved active sites and Ag (100) facets that preferentially stabilize the *CO intermediate.

## Figures and Tables

**Figure 1 molecules-30-03475-f001:**
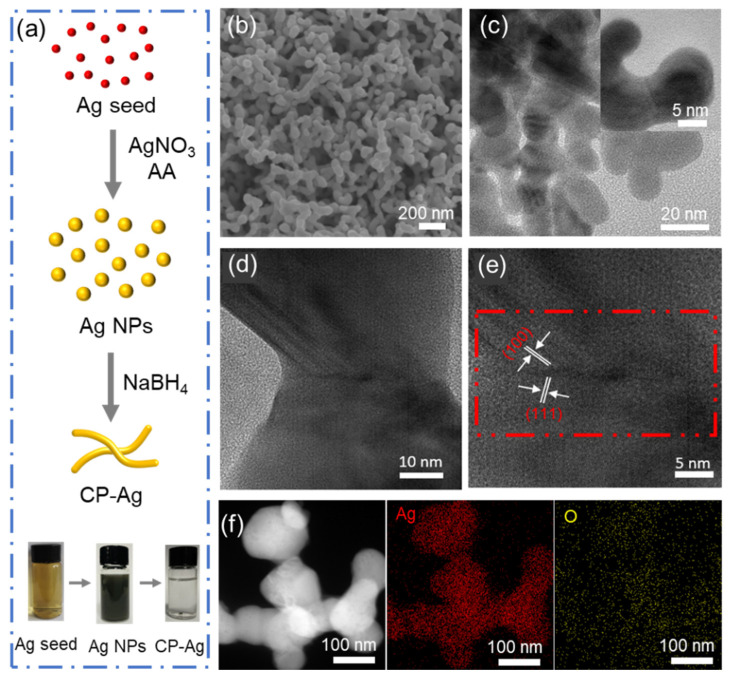
(**a**) Schematics of the synthesis route; (**b**) the SEM images of CP-Ag; (**c**–**e**) TEM image of CP-Ag; The red dotted line represents the area with grain boundary. (**f**) HAADF and EDS mapping images of CP-Ag.

**Figure 2 molecules-30-03475-f002:**
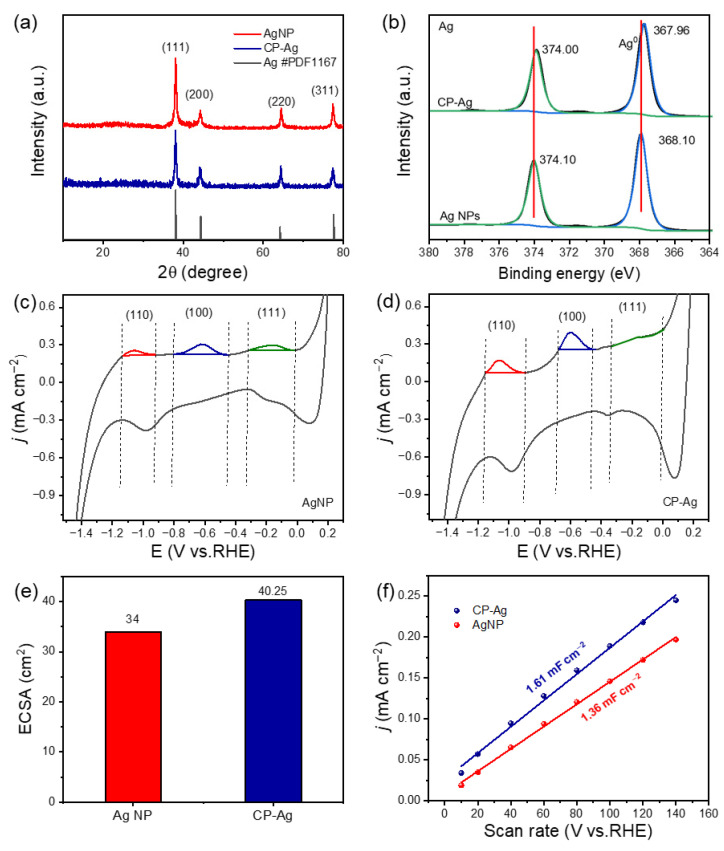
(**a**) XRD patterns of AgNP and CP-Ag; (**b**) XPS spectra of AgNP and CP-Ag for Ag 3d; Cyclic voltammograms at a scan rate of 50 mV/s for (**c**) AgNP and (**d**) CP-Ag; The peak intensities of red, blue, and green represent Ag (110), (100), and (111) crystal planes, respectively. (**e**) ECSA plots of CP-Ag and AgNP; (**f**) Cdl plots of CP-Ag and AgNP.

**Figure 3 molecules-30-03475-f003:**
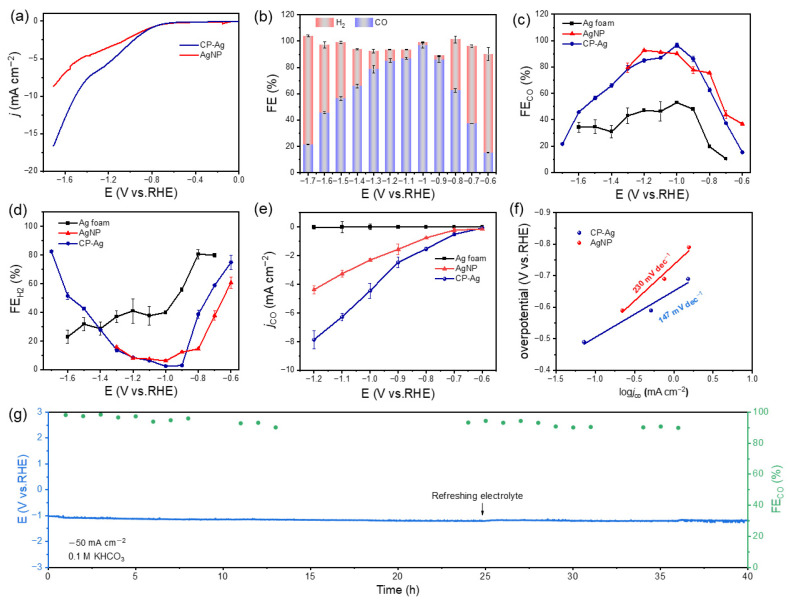
(**a**) A comparison of current density versus potential; (**b**) the FE of CO and H_2_ for CP-Ag; (**c**) the FE of CO for CP-Ag, AgNP and Ag foam; (**d**) the FE of H_2_ for CP-Ag, AgNP and Ag foam; (**e**) CO partial current density of CP-Ag and AgNP; (**f**) overpotential versus CO production partial current density AgNP and CP-Ag; (**g**) the long-term stability test of CP-Ag in GDE-based flow cell with 0.1 M KHCO_3_.

**Figure 4 molecules-30-03475-f004:**
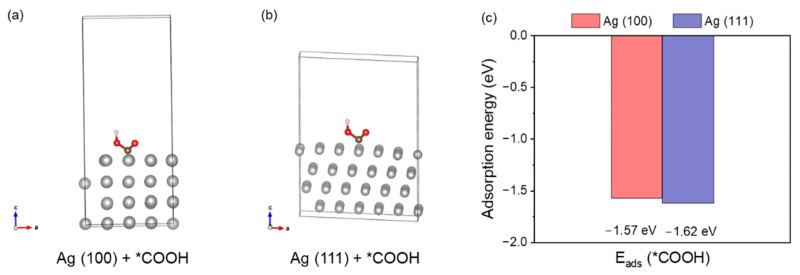
(**a**) The adsorption configuration of Ag (100) + *COOH; (**b**) the adsorption configuration of Ag (111) + *COOH; (**c**) the E_ads_(*COOH) of Ag (100) and Ag (111) (Ag: gray).

**Table 1 molecules-30-03475-t001:** Comparison of electrochemical properties between CP-Ag and AgNPs on different crystal facets.

Material	CP-Ag			AgNPs		
Crystal Facet	110	100	111	110	100	111
Area (×10^−3^ cm^2^)	4.1	12.3	6.3	11.1	15.8	0.148
Capacitance (mF)	0.43	0.66	0.02	0.20	0.34	0.22
Proportion (%)	38.8	59.5	1.7	25.9	44.8	28.3

## Data Availability

All data generated or analyzed during this study are included in the published article and its Appendix A.

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
