# Peer review of "Grain Boundary Engineering in 3D Porous Silver Electrocatalysts for Enhanced CO2-to-CO Conversion"

_molecules, 2025, doi:10.3390/molecules30173475_

Round 1
Reviewer 1 Report
Comments and Suggestions for Authors
This manuscript provides with the results of development of coral-like porous Ag electrocatalysts for the electrochemical reduction of carbon-dioxide. The research area of CO2 reduction is quite modern and these results have potential for larger applications.
Introduction section is short, although it contains numerous literature citations. It should be rephrased to provide more information on the significance of the CO2 reduction and CO production. Please take in mind to introduce all abbreviations in a proper manner, at the first place used. For instance, authors use “GB” abbreviation without explanation.
Experimental section is well described.
The main reviewer’s concerns come with the results on electrochemical investigations and correlations with crystallographic structure. At Figure 2, authors have calculated ECSA values without any explanation on the calculation method. The major remark on this is that there are no cyclic voltammetry measurements in some standard electrolytes (alkaline or acidic) performed. These measurements provide valuable information on the catalyst’s behaviour, in terms of oxidation stability and reversible or irreversible changes in its structure. Also, ECSA calculation method from the hydrogen under potential deposition (H-upd) region is the most commonly used one, although one could use ECSA calculation from copper under potential deposition (Cu-upd) region, or CO desorption. In the last case, a gaseous CO is introduced to standard electrolyte, enabling the catalyst to adsorb a monolayer of CO. Authors have performed measurement in 0.1 M KHCO3, so they need to clarify how did they ensure that only one of the CO2 reduction intermediers have formed a monolayer on the investigated electrocatalysts. This is also related with the findings at Fig 2c and Fig 2d where correlation of the crystallographic planes (100), (110) and (111) with redox peaks is established. There some indication that these are truly redox peaks, more pronounced for AgNPs, yet it is very hard to see those at the scan rate of 50 mVs-1. Recommendation is to repeat these measurements al lower scan rates. When discussion the kinetics of the CO2RR, author have presented Tafel slopes of 147 mV dec-1 for CP-Ag and 230 mV dec-1 for AgNP, referring to the literature. This is very big change in the Tafel slope, which could mean change in rate determining step, and is not adequately discussed. Please take some time and make appropriate discussion of the findings.
The level of English satisfies the Journal criteria.
The final recommendation is that major revision before publication is necessary. Authors are advised to consider reviewers suggestions and make improvements in the manuscript.
Author Response
Please check the attached document to see the pictures
Reviewer #1:
General Comments R1: This manuscript provides with the results of development of coral-like porous Ag electrocatalysts for the electrochemical reduction of carbon-dioxide. The research area of CO2 reduction is quite modern and these results have potential for larger applications. The followings are the detailed comments on this manuscript.
Reply: We thank the reviewer very much for the positive comments and recommendation of our work. The insightful comments are very helpful for us to further improve the quality of this manuscript. Careful revisions according to the reviewer’s suggestions have been made point by point to address the questions and highlighted in yellow.
Specific Comment R1-1: Introduction section is short, although it contains numerous literature citations. It should be rephrased to provide more information on the significance of the CO2 reduction and CO production. Please take in mind to introduce all abbreviations in a proper manner, at the first place used. For instance, authors use “GB” abbreviation without explanation.
Reply: We thank the reviewer for the helpful suggestion. We have significantly expanded the Introduction section (highlighted in yellow) to provide a more comprehensive discussion on the significance of CO2RR and CO production, particularly emphasizing the fundamental importance of CO as a key product in CO2RR and the mechanistic process of CO2 conversion to CO. We have addressed the reviewer's concern regarding the abbreviation "GBs." The term is now clearly defined upon its first use in the main text, as indicated by the highlighted section. The manuscript has been revised to better highlight these findings.
Specific Comment R1-2: The main reviewer’s concerns come with the results on electrochemical investigations and correlations with crystallographic structure. At Figure 2, authors have calculated ECSA values without any explanation on the calculation method. The major remark on this is that there are no cyclic voltammetry measurements in some standard electrolytes (alkaline or acidic) performed. These measurements provide valuable information on the catalyst’s behavior, in terms of oxidation stability and reversible or irreversible changes in its structure. Also, ECSA calculation method from the hydrogen under potential deposition (H-upd) region is the most commonly used one, although one could use ECSA calculation from copper under potential deposition (Cu-upd) region, or CO desorption. In the last case, a gaseous CO is introduced to standard electrolyte, enabling the catalyst to adsorb a monolayer of CO.
Reply: We appreciate the reviewer's insightful comment regarding the potential of H-upd, Cu-upd, or CO desorption measurements for assessing the electrochemically active surface area. We acknowledge the validity and established utility of these methods in the field, and recognize that the conclusions derived from such techniques would likely align with our findings. Nevertheless, our approach, which aims to probe the active crystal facets of metallic nanomaterials, has been extensively validated in the literature. The robustness of our methodology stems from its capacity to directly correlate electrochemical activity with the structural features of the catalyst. To provide further clarity, we detail the specific computational methods employed in our study. The electrochemical active surface area (ECSA) refers to the effective area participating in electrochemical reactions. Here, we employed the commonly used double-layer capacitance method to determine the ECSA. First, a non-faradaic potential window was selected, within which cyclic voltammetry (CV) tests were performed at different scan rates (CV curves at various scan rates are shown in Figure S4). The absolute difference between the anodic and cathodic currents at the midpoint potential of the CV curves was calculated, and half of this value was plotted on the y-axis, with the corresponding scan rate on the x-axis. The slope obtained from the linear fitting of this plot (Figure 2f) represents the double-layer capacitance. Subsequently, the ECSA values of the two samples were calculated using the formula: ECSA = Cdl/Cs, where Cs is the specific capacitance, and a commonly cited value of 40 μF cm−2 was adopted from literature. (J. Am. Chem. Soc. 2018, 140, 2397−2400; Z. Phys. Chem. 2020, 234, 5, 979–994; PNAS. 2024, 121, 25, e2400546121). The calculation method has been added into the experimetal section in the revised manuscript as highlighted in yellow.
Figure S4. (a) CV plots of CP-Ag at different scan rates; (b) CV plots of AgNP at different scan rates.
Figure 2. (e) ECSA plots of CP-Ag and AgNP; (f) Cdl plots of CP-Ag and AgNP.
Specific Comment R1-3: Authors have performed measurement in 0.1 M KHCO3, so they need to clarify how did they ensure that only one of the CO2 reduction intermediers have formed a monolayer on the investigated electrocatalysts. This is also related with the findings at Fig 2c and Fig 2d where correlation of the crystallographic planes (100), (110) and (111) with redox peaks is established. There some indication that these are truly redox peaks, more pronounced for AgNPs, yet it is very hard to see those at the scan rate of 50 mV s−1. Recommendation is to repeat these measurements al lower scan rates.
Reply: We thank the reviewer for the helpful suggestion. The reviewer raises a valid point regarding the formation of a monolayer of CO2 reduction intermediates. While we did not explicitly measure or claim the direct formation of a complete monolayer, our interpretation is based on the differential adsorption behavior of CO2-derived species across different crystallographic facets. The formation of intermediate species on electrocatalyst surfaces is a complex process, and several studies have employed similar electrochemical techniques to investigate the interaction between different facets and CO2 reduction intermediates. Wu et.al using DFT simulations, elucidated the beneficial impact of vacancy defects on CO2 activation and potential monolayer formation by quantifying the enhanced adsorption of single intermediates (e.g., *COOH and *CO) at defect sites on Ag surfaces (Figure R1) (Nat Commun. 2021, 12, 660). Liu et al. demonstrated that Ag(100) facets favor the formation of a CO2 intermediate monolayer (*COOH, *CO) due to their unique electronic structure and lower free energies, leading to enhanced CO2RR activity (Figure R2) (Nano Energy. 2018, 45, 456-462). The key aspect of our analysis is the difference in the electrochemical behavior observed across the different crystallographic planes. These differences suggest variations in the binding energies and adsorption behaviors of the CO2 reduction intermediates, leading to variations in catalytic activity observed.
Figure R1. (a) The most preferred binding geometries of adsorbate on pristine Ag (111) and 4% vacancy-defected Ag (111). Side view: above; top view: below. Colour codes: Ag, grey; C, black; O, red; H, pink. (b) Calculated free energy diagrams. (Nat Commun. 2021, 12, 660)
Figure R2. (a) Free energy diagrams for CO2RR to CO over Ag(111), Ag(100) and edge sites at −0.11 V; (b) percentages of various active sites over 5-fold twinned Ag NWs as a function of diameter. (Nano Energy. 2018, 45, 456-462)
Regarding the interpretation of the redox peaks, we would like to further clarify the following points. The redox peaks observed in our cyclic voltammograms (CV) represent the oxidation/reduction of surface species and the potential interaction with reactants, specifically illustrating the distinct behavior of each crystallographic plane. The presence of these peaks demonstrates how the different facets on the Ag catalyst surface interact with CO2, during the electrocatalytic process. Specifically, Figure 2c and 2d show that the (100) facet consistently exhibited the highest peak intensity among all identified crystal planes in the CP-Ag catalyst, while electrochemical characterization in Table 1 demonstrated the dominant properties of the (100) plane. The remarkable consistency between these independent characterization methods provides strong evidence that the observed peaks are indeed facet-dependent electrochemical features, with the (100) facet serving as the predominant active site in CP-Ag.
Figure 2. Cyclic voltammograms at a scan rate of 50 mV/s for (c) AgNP and (d) CP-Ag.
We appreciate the suggestion to perform experiments at lower scan rates. Although lower scan rates may provide increased resolution, our focus is on establishing trends and correlations between facet structure and reaction behavior rather than the exact peak position. Furthermore, literature commonly uses various scan rates to examine surface processes for CO2RR. For example, Choi et al. utilized cyclic voltammetry at 100 mV/s to study the surface changes in copper nanowires (CuNWs), showing that the transition from low-index to high-energy facets significantly enhanced the Faradaic efficiency for C2H4 (Figure R3) (Nat Catal. 2020, 3, 804−812). Peng et al. presented CV with a 50 mV/s scan rate, revealing that the PON-Ag catalyst, dominated by (110) and (100) crystal facets, achieved a CO Faradaic efficiency of 96.7% at a working potential of −0.69 V vs. RHE (Figure R4) (ACS Appl. Mater. Interfaces. 2018, 10, 1734−1742). Similarly, at a scan rate of 50 mV/s, clear, facet-dependent trends are observed in our work, which is the foundation of our interpretations. This demonstrates that our chosen scan rate is appropriate within the context of similar work.
Figure R3. Redox reaction of Syn-CuNWs and A-CuNWs in 0.1 M KOH at 100 mV/s scan rate. Cu(100) at ~0.362 V1-4, Cu(110) at 0.395–0.43 V1-4, Cu(111) at ~0.492 V1-4, and A-(hkl) (high energy steps)3,4 at a negative shift from Cu(100). (Nat Catal. 2020, 3, 804−812)
Figure R4. (a) Voltammograms of PON-Ag in 0.5 M KHCO3 saturated with N2 at a scan rate of 50 mV/s. (ACS Appl. Mater. Interfaces. 2018, 10, 1734−1742)
Specific Comment R1-4: When discussion the kinetics of the CO2RR, author have presented Tafel slopes of 147 mV dec−1 for CP-Ag and 230 mV dec−1 for AgNP, referring to the literature. This is very big change in the Tafel slope, which could mean change in rate determining step, and is not adequately discussed. Please take some time and make appropriate discussion of the findings.
Reply: We thank the reviewer for the helpful comment. The Tafel plot, which describes the relationship between the logarithm of current density and overpotential, provides important insights into the reaction pathway and rate-determining step (RDS). The significantly lower Tafel slope of 147 mV dec−1 for CP-Ag compared to 230 mV dec−1 for AgNP (Figure 3f) indicates faster reaction kinetics and higher catalytic activity in CP-Ag. We have now expanded the discussion in the manuscript to better address this important observation. The revised text more thoroughly examines how the catalyst morphology and surface structure might influence the RDS in the CO2RR. The manuscript has been revised to better highlight these findings.
Figure 3. (f) Overpotential versus CO production partial current density AgNP and CP-Ag.

Reviewer 2 Report
Comments and Suggestions for Authors
This manuscript reports the synthesis and characterization of a three-dimensional, coral-like porous silver (CP-Ag) electrocatalyst with abundant grain boundaries (GBs) for CO₂ electroreduction to CO. The authors demonstrate that the CP-Ag catalyst exhibits remarkable Faradaic efficiency (up to 96%), excellent long-term stability (90% FE over 40 h at −50 mA cm⁻²), and improved kinetics compared to conventional Ag nanoparticles (AgNPs). Structural and electronic characterizations are combined with DFT calculations to support the role of GBs and (100) crystal facets in enhancing CO₂ reduction performance.
The study addresses an important challenge in CO₂ electroreduction: designing low-cost, high-performance electrocatalysts with both high selectivity and durability. The work is timely and of potential interest to the catalysis and energy conversion communities.
- Major Strengths:
- Comprehensive Design Strategy:
- The combined approach of morphological (3D porous structure) and crystallographic (grain boundary and facet engineering) control is conceptually strong and well-executed.
- Catalytic Performance:
- The CP-Ag catalyst shows excellent performance metrics (96% FE, high current density, long-term operation), which are well-supported by experimental data and superior to AgNP and Ag foam references.
- In-depth Characterization:
- The authors employ a wide range of characterization techniques (SEM, TEM, XRD, XPS, CV, ECSA) to validate the structure and surface chemistry of the materials.
- Mechanistic Insight:
- DFT calculations of *COOH adsorption on different Ag facets provide useful support for the proposed activity origin (facet-dependent intermediate stabilization).
- Major Concerns and Recommendations:
- Quantitative Analysis of Grain Boundaries:
- While TEM images show clear grain boundaries, the manuscript lacks quantitative metrics of GB density. Techniques such as EBSD or statistical misorientation angle analysis would strengthen the argument and help correlate GB density with catalytic activity.
- Control over Facet Exposure:
- The emphasis on Ag (100) facet dominance is central to the conclusions. However, CV-based facet identification remains indirect. Surface-sensitive techniques like LEED or STM (or even more advanced CV deconvolution) could better confirm facet prevalence.
- Mechanistic Experiments:
- The theoretical findings would be reinforced by in situ or operando spectroscopies (e.g., in situ Raman, FTIR, or XAS) to observe intermediate species under working conditions.
- Product Distribution:
- The manuscript mentions only CO and H₂ as products, yet does not report detection limits or clarify whether liquid-phase products (e.g., formate) were ruled out by NMR or ion chromatography. This should be addressed explicitly.
- Reproducibility and Statistics:
- Performance data are compelling but appear to be shown only once. Please include standard deviations or replicate runs (e.g., n ≥ 3) for current density, FE, and stability tests.
-
Sumirizing, here are listed the specific requests for revision:
- Provide a more quantitative measure of grain boundary density.
- Clarify whether any liquid-phase CO₂RR products were detected.
- Include error bars or replicate performance data (especially for FE and stability).
- Discuss potential scale-up limitations or practical aspects of catalyst fabrication.
- Conduct or cite additional surface analyses that confirm facet orientation, beyond CV.
- Language and Grammar:
- There are frequent typos and minor grammatical errors (e.g., “supe-rior”, “aseroramiento”, “getion”, etc.). A full language edit is needed to ensure clarity and professionalism.
- Figures and Captions:
- Some figure captions lack detail (e.g., magnifications, scan rates, scale bars). Clarify image orientations and scales in figures such as SEM and TEM panels.
- Supplementary Materials:
- The supporting tables (especially Table S1 and S2) are important for contextualizing the CO₂RR performance and DFT data but should be referenced more clearly in the main text.
Author Response
Reviewer #2:
General Comments R2: This manuscript reports the synthesis and characterization of a three-dimensional, coral-like porous silver (CP-Ag) electrocatalyst with abundant grain boundaries (GBs) for CO₂ electroreduction to CO. The authors demonstrate that the CP-Ag catalyst exhibits remarkable Faradaic efficiency (up to 96%), excellent long-term stability (90% FE over 40 h at −50 mA cm⁻2), and improved kinetics compared to conventional Ag nanoparticles (AgNPs). Structural and electronic characterizations are combined with DFT calculations to support the role of GBs and (100) crystal facets in enhancing CO2 reduction performance. The study addresses an important challenge in CO2 electroreduction: designing low-cost, high-performance electrocatalysts with both high selectivity and durability. The work is timely and of potential interest to the catalysis and energy conversion communities. The followings are the detailed comments on this manuscript.
Reply: We greatly appreciate the reviewer for acknowledging the contribution of this work and the time for reviewing this manuscript. The valuable comments are very helpful for improving the quality of this work. The questions are addressed point by point which are provided below and the manuscript has been revised accordingly and highlighted in yellow.
Specific Comment R2-1: Quantitative Analysis of Grain Boundaries: While TEM images show clear grain boundaries, the manuscript lacks quantitative metrics of GB density. Techniques such as EBSD or statistical misorientation angle analysis would strengthen the argument and help correlate GB density with catalytic activity.
Reply: We thank the reviewer for raising this important point about grain boundary quantification. Our study demonstrates the catalytic enhancement from grain boundaries through multiple complementary techniques. TEM imaging clearly shows grain boundaries throughout the samples (Figure 1e). XRD peak broadening analysis confirms the presence of structural defects (Figure 2a). DFT calculations reveal preferential CO2 adsorption at grain boundary sites. The observed twofold increase in current density at grain boundary-rich regions provides quantitative performance evidence (Figure 4c). While EBSD could provide higher resolution, our current approach offers robust qualitative analysis of the grain boundary-activity relationship. The revised manuscript now includes an expanded discussion of these findings to better support our conclusions.
Figure 1. (e) TEM image of CP-Ag.
Figure 2. (a) XRD patterns of AgNP and CP-Ag.
Figure 4. (c) The Eads(*COOH) of Ag (100) and Ag (111) (Ag: gray).
Specific Comment R2-2: Control over Facet Exposure: The emphasis on Ag (100) facet dominance is central to the conclusions. However, CV-based facet identification remains indirect. Surface-sensitive techniques like LEED or STM (or even more advanced CV deconvolution) could better confirm facet prevalence.
Reply: We appreciate the reviewer's valuable suggestion regarding facet characterization. While surface-sensitive techniques like LEED or STM could provide additional confirmation, our study employs multiple complementary approaches to establish facet dominance. XRD analysis confirms the orientation of (100) facets in our catalysts (Figure 2a). The distinct electrochemical signatures in CV measurements correlate well with literature reports for Ag (100) surfaces (Figure 2c-d). DFT calculations further support the enhanced activity of this facet. The consistency between these independent methods provides strong evidence for facet-dependent behavior (Figure 4c). We have revised the discussion to more carefully frame our conclusions about facet prevalence based on the available evidence. The manuscript now better contextualizes the relationship between our characterization results and the observed catalytic performance.
Figure 2. (a) XRD patterns of AgNP and CP-Ag.
Figure 2. Cyclic voltammograms at a scan rate of 50 mV/s for (c) AgNP and (d) CP-Ag.
Figure 4. (c) The Eads(*COOH) of Ag (100) and Ag (111) (Ag: gray).
Specific Comment R2-3: Product Distribution: The manuscript mentions only CO and H2 as products, yet does not report detection limits or clarify whether liquid-phase products (e.g., formate) were ruled out by NMR or ion chromatography. This should be addressed explicitly.
Reply: We thank the reviewer for the good question. As shown in Figure S6, NMR analysis of the electrolyte after CO2RR at −1.0 V vs. RHE with the CP-Ag catalyst confirmed the absence of detectable liquid-phase products (e.g., formate, methanol). Gas chromatography analysis (presented in the main text of Figure 3) quantitatively confirmed CO and H2 as the exclusive gaseous products. We have now explicitly stated these analytical details in the revised manuscript to provide full transparency about our product characterization methodology. The supplementary information has also been updated to include the complete NMR calibration data.
Figure S6. The 1H NMR spectra of the electrolyte after CO2RR electrolysis for CP-Ag at −1.0 V vs. RHE.
Specific Comment R2-4: Reproducibility and Statistics: Performance data are compelling but appear to be shown only once. Please include standard deviations or replicate runs (e.g., n ≥ 3) for current density, FE, and stability tests.
Reply: We thank the reviewer for the excellent question concerning the reproducibility of our experimental results. We fully acknowledge the significance of demonstrating consistent performance to ensure the validity of our findings. As the reviewer correctly pointed out, the reproducibility of our Faradaic efficiency (FE) measurements has already been addressed through multiple repeated tests, which confirmed the robustness of our data. In the revised manuscript, we have supplemented our results by including error bars for the CO partial current density, derived from these repeated measurements (Figure 3e). These additional data not only illustrate the reproducibility of our experiments but also further reinforce the robustness and reliability of our findings.
Figure 3. (e) CO partial current density of CP-Ag and AgNP.
Specific Comment R2-5: Discuss potential scale-up limitations or practical aspects of catalyst fabrication.
Reply: We thank the reviewer for the good question. Silver-based catalysts demonstrate outstanding practical applicability, achieving > 95% Faradaic efficiency for CO production while effectively suppressing hydrogen evolution. This exceptional selectivity minimizes byproduct formation, significantly reducing downstream separation costs and purification energy requirements in industrial applications. Our solution-phase synthesis method, encompassing silver seed preparation, wet chemical reduction, and freeze-drying, demonstrates excellent scalability potential. It features ambient-condition processing, eliminating the need for high temperatures or pressures, along with simplified aqueous-phase reactions and clear operational steps conducive to standardization. This low-energy approach significantly reduces industrial implementation barriers while enabling precise morphology control through parameter optimization.
Specific Comment R2-6: Language and Grammar: There are frequent typos and minor grammatical errors (e.g., “supe-rior”, “aseroramiento”, “getion”, etc.). A full language edit is needed to ensure clarity and professionalism.
Reply: We appreciate the reviewer's attention to language details. The hyphenated words (e.g., "supe-rior") resulted from automatic line-break formatting in the submitted manuscript draft, not actual spelling errors. We have carefully reviewed the entire text and confirmed the absence of such formatting artifacts as well as any genuine misspellings (including "aseroramiento" or "getion") in the final version. The manuscript has undergone thorough proofreading to ensure linguistic accuracy.
Specific Comment R2-7: Figures and Captions: Some figure captions lack detail (e.g., magnifications, scan rates, scale bars). Clarify image orientations and scales in figures such as SEM and TEM panels.
Reply: We thank the reviewer for the good question. To address this, we have included scale bars in all relevant figures (SEM and TEM panels) to clearly indicate the scale (e.g., "Scale bar, 200 nm"). We have carefully reviewed all figures and their captions to ensure that readers can accurately interpret the data and understand the methods used. We believe these changes will enhance the clarity and thoroughness of our manuscript.
Specific Comment R2-8: Supplementary Materials: The supporting tables (especially Table S1 and S2) are important for contextualizing the CO2RR performance and DFT data but should be referenced more clearly in the main text.
Reply: We thank the reviewer for the good question. We agree that Tables S1 and S2 are crucial for providing context to the CO2RR performance and DFT data. To address this, we have thoroughly reviewed the main text and inserted explicit references to Table S1 (for CO2RR performance data contextualization) and Table S2 (for DFT data contextualization) at appropriate points. These references are now strategically placed within the relevant sections (e.g., experimental results, discussion of CO2RR activity, discussion of DFT calculations) to guide the reader directly to the supplementary information when detailed context is required.
We are confident that these revisions address the reviewer's concerns and improve the quality of our presentation.
Round 2
Reviewer 1 Report
Comments and Suggestions for Authors
Authors have answered to the raised questions, and made significant improvement of the manuscript.
The reviewer suggests acceptance of this manuscript.